# Neogrisphenol A, a Potential Ovarian Cancer Inhibitor from a New Record Fungus *Neohelicosporium griseum*

**DOI:** 10.3390/metabo13030435

**Published:** 2023-03-16

**Authors:** Li-Juan Zhang, Ming-Fei Yang, Jian Ma, Xing-Juan Xiao, Xiao-Yan Ma, De-Ge Zheng, Mei-Yan Han, Ming-Lei Xia, Ruvishika S. Jayawardena, Ausana Mapook, Yuan-Pin Xiao, Ji-Chuan Kang, Yong-Zhong Lu

**Affiliations:** 1School of Food and Pharmaceutical Engineering, Guizhou Institute of Technology, Guiyang 550003, China; 6471105010@lamduan.mfu.ac.th (L.-J.Z.); xiaoxingjuanbenjuan@gmail.com (X.-J.X.); 20120065@git.edu.cn (X.-Y.M.); degezheng@gmail.com (D.-G.Z.); meiyanhan04@gmail.com (M.-Y.H.); xml0312xml@gmail.com (M.-L.X.); emaypx@gmail.com (Y.-P.X.); 2Center of Excellence in Fungal Research, Mae Fah Luang University, Chiang Rai 57100, Thailand; 6371105007mj@gmail.com (J.M.); ruvishika.jay@mfu.ac.th (R.S.J.);; 3School of Science, Mae Fah Luang University, Chiang Rai 57100, Thailand; 4Engineering and Research Center for Southwest Bio-Pharmaceutical Resources of National Education Ministry of China, Guizhou University, Guiyang 550025, China; mf_yang@126.com; 5College of Life Sciences, Guizhou University, Guiyang 550025, China; 6Department of Health Management, Guiyang Healthcare Vocational University, Guiyang 550081, China

**Keywords:** saprophytic fungi, helicosporous hyphomycetes, polyketide derivatives, antimicrobials, cytotoxicity

## Abstract

From the rice fermentation product of a new record fungus, *Neohelicosporium griseum*, two new polyketides, neogrisphenol A (**1**) and neogrisphenol B (**2**), one new isochroman-1-one, (*S*)-6-hydroxy-7-methoxy-3,5-dimethylisochroman-1-one (**3**), and four known compounds (**4**–**7**) were isolated. Their structures were determined using 1D- and 2D-NMR, mass spectrometry, and chemical calculations. The C-3~C-2′ polymerization mode between the two *α*-naphthalenone derivative moieties is uncommon in compounds **1** and **2**. Meanwhile, compounds **1**–**2** and **5** exhibited antibacterial activity against *Bacillus subtilis*, *Clostridium perfringens*, *Staphylococcus aureus*, and *Staphylococcus aureus*, with MIC values ranging between 16 and 31 µg/mL. In addition, compound **5** showed antifungal activity against *Sclerotinia sclerotiorum* and *Phytophthora nicotianae var. nicotianae*, with respective IC_50_ values of 88.14 ± 2.21 µg/mL and 52.36 ± 1.38 µg/mL. Compound **1** showed significant cytotoxicity against A2780, PC-3, and MBA-MD-231 cell lines with respective IC_50_ values of 3.20, 10.68, and 16.30 µM, and the cytotoxicity against A2780 cells was even higher than that of cisplatin (CDDP). With an IC_50_ value of 10.13 µM, compound **2** also exhibited cytotoxicity against A2780. The in vitro results showed that compound **1** inhibited A2780 cell proliferation, induced apoptosis, and arrested the cell cycle at the S-phase in a concentration-dependent manner.

## 1. Introduction

Secondary metabolites of fungi have enormous potential, particularly for drug discovery [1]. More than half of the compounds isolated from fungi were antibacterial, antifungal, or antitumor [1]. Research on secondary metabolites of fungi concentrated primarily on endophytic fungi of medicinal plants, whereas research on secondary metabolites of saprophytic fungi was relatively uncommon [2].

Helicosporous hyphomycetes, such as the species of *Helicosporium*, *Helicoma*, *Helicomyces,* and *Tubeufia,* are capable of producing a variety of bioactive compounds [3,4,5,6,7,8,9]. The vast majority of helicosporous hyphomycetes are members of the family *Tubeufiaceae* (Tubeufiales, Dothideomycetes) [10]. However, this family study focuses primarily on morphology and phylogeny [10,11], and reports on secondary metabolites are scarce. 

In this study, bioassay-guided isolation led to the discovery of three new compounds [two new polyketide derivatives, neogrisphenol A (**1**) and neogrisphenol B (**2**), and a new isochroman-1-one derivative named (*S*)-6-hydroxy-7-methoxy-3,5-dimethylisochroman-1-one (**3**)] and four known compounds from the EtOAc extracts of the rice fermentation product of a new record fungus, *Neohelicosporium griseum*. Compounds **1** and **2** have a rare C-3~C-2′ linkage between the cyclohexanone ring of one *α*-naphthalone derivative and the cyclohexanone ring of another *α*-naphthalone derivative, which has not been reported in natural or synthetic products. The antimicrobial and cytotoxic activities of compounds **1**–**7** were evaluated. Compound **1** exhibited significant cytotoxicity against the A2780 cell line with IC_50_ values of 3.20 µM, higher than that of cisplatin (CDDP). The results showed that this compound might serve as a potential inhibitor of ovarian cancer.

## 2. Materials and Methods

### 2.1. General Experimental Procedure

Ultraviolet-visible (UV/Vis) spectra were acquired using a UV-Vis spectrophotometer, the UV-5300 (Hitachi, Tokyo, Japan). Infrared (IR) spectra were measured on a Bruke Vertex 80 (KBr disks) infrared spectrometer. Optical rotations were recorded in methanol (MeOH) solution on an AUTOPOL1 polarimeter at 28 °C (Rudolph, Wilmington, MA, USA). CD spectra were obtained on a J-810 circular dichroism spectropolarimeter (JASCO Corp., J-810, Tokyo, Japan). Electro-spray ionization mass spectrometry (ESIMS) and high-resolution electrospray ionization mass spectrometry (HRESIMS) analyses were measured on a Waters Xevo TQS and a Waters Xevo G2S Q-TOF LC/MS, respectively. NMR spectra were recorded on a Bruker 600 MHz instrument, and tetramethylsilane (TMS) was used as an internal standard. Column chromatography was performed on silica gel (200−300 mesh, Qingdao Marine Chemical Co., Ltd., Qingdao, China) and Sephadex LH-20 (Amersham Biosciences, Uppsala, Sweden), respectively. HPLC analysis was performed on an Agilent 1260 with a C18 column (Agilent Extend-C18, 4.6 × 250 mm, 5 μm, 1 mL/min). HPLC separation was performed on Shimadzu Soviet production LC-16P with an ODS column (Silgreen C18AB, 10 × 250 mm, 5 μm, 180 Å, 3 mL/min). Cell apoptosis and cycle kits were purchased from BD Biosciences (San Jose, CA, USA).

### 2.2. Fungal Taxonomy and Identification

Decaying wood samples were collected randomly from sites in Guiyang Forest Park, China. The specimen examination, micromorphological study, DNA extraction, PCR amplification, sequencing, and phylogenetic analysis were conducted using the method described in Lu et al. [12]. 

### 2.3. Fermentation, Extraction, and Isolation

The strain of *Neohelicosporium griseum* was cultured on PDA at 28 °C for 10 days and then was inoculated into 20 × 250 mL Erlenmeyer flasks, each containing a 125 mL liquid medium (maltose 20 g, sodium glutamate 10 g, potassium dihydrogen phosphate 0.5 g, magnesium sulfate 0.3 g, glucose 10 g, yeast powder 3 g, mannitol 20 g, tap water 1 L). These flasks were incubated on a shaking table at 28 °C and 180 RPM for 7 days. About 5 mL of the seed liquid was transferred to a 200-mL plastic bag prepared from 50 g rice and 55 mL of distilled water. These bags were incubated at 28 °C under static conditions for 3 months. A total of 400 bags were fermented. 

The fermented product was extracted three times with ethyl acetate (EtOAc)/methanol (10:1) (each 50 L), and the combined EtOAc/methanol (10:1) solutions were dried in vacuo to yield 99.4 g of crude extract. The 99.4 g of crude extract was subjected to column chromatography (CC) over silica gel (PE/EtOAc, *v*/*v*, 100:0→0:100) to give 20 fractions (Fr.1–20). Of these, Fr.12 was separated with Sephadex LH-20 in MeOH to yield 4 subfractions (Fr.12.1–12.4). Sephadex LH-20 was used to separate Fr.12.4 into two subfractions (Fr.12.4.1–12.4.2) in MeOH. Fr.12.4.2 was further purified by RP-HPLC with MeOH/H_2_O (50:50, 3 mL/min) to afford 4 (*t*_R_ = 18.5 min, 3.0 mg). Fr.12.3 was chromatographed again with silica gel (PE/EtOAc, *v*/*v*, 7:3) to afford 4 subfractions (Fr.12.3.1–12.3.4). Fr.12.3.2 was further purified by RP-HPLC with MeOH (3 mL/min) to afford 7 (*t*_R_ = 13.0 min, 6.3 mg). Fr.15 was divided into 4 subfractions (Fr.15.1–15.4) using Sephadex LH-20 in MeOH. Fr15.3 was further purified by RP-HPLC with MeOH/H_2_O (60:40, 3 mL/min) to afford 1 (*t*_R_ = 10.3 min, 140.0 mg) and 5 (*t*_R_ = 13.4 min, 12.4 mg). Seven subfractions (Fr.18.1–18.7) of Fr.18 were obtained after separation using Sephadex LH-20 in MeOH. Fr18.2 was further purified by RP-HPLC with MeOH/H_2_O (35:65, 3 mL/min) to afford 3 (*t*_R_ = 16.6 min, 16.5 mg). Fr.18.4 was further separated with Sephadex LH-20 in CHCl_2_/MeOH (1:1) to yield 3 subfractions (Fr.18.4.1–18.4.3). Fr.18.4.3 was further purified by RP-HPLC with MeOH/H_2_O (40:60, 3 mL/min) to afford 6 (*t*_R_ = 18.2 min, 91.0 mg). Fr18.7 was further purified by RP-HPLC with MeOH/H_2_O (40:60, 3 mL/min) to afford 2 (*t*_R_ = 22.0 min, 6.0 mg).

### 2.4. Biological Assays

#### Antimicrobial and Cytotoxic Activities of Compounds **1**–**7**

Compounds were evaluated for their antimicrobial activity against two fungi (*Candida albicans* and *C. glabrata*), three agricultural pathogenic fungi (*Fusarium graminearum*, *Phytophthora nicotianae var. nicotianae*, *Sclerotinia sclerotiorum*), five Gram-positive bacteria (*Bacillus subtilis*, *Clostridium perfringens*, *Ralstonia solanacearum*, *Staphylococcus aureus*, MRSA strain *Staphylococcus aureus*), and three Gram-negative bacteria (*Escherichia coli, Paratyphoid fever, Pseudomonas aeruginosa*), using ciprofloxacin as a positive control against Gram-positive and Gram-negative bacteria, while ketoconazole was used as an antifungal positive control. In addition, the compounds’ cytotoxicities against three mammalian cell lines (human ovarian cancer A2780, human prostate cancer PC-3, and human breast cancer MBA-MD-231 cell lines) were examined by the MTT method, using cisplatin (CDDP) as the positive control against A2780 and PC-3, while adriamycin was used as a positive control against MBA-MD-231. The A2780 cell line was purchased from Xiamen Immocell Biotechnology Co., Ltd. (Xiamen, China). The PC-3 cell line was purchased from Shanghai Zhongqiao Xinzhou Biological Technology Co., Ltd. (Shanghai, China). The MBA-MD-231 cell line was purchased from Cell bank of Chinese Academy of Sciences (Shanghai, China). Following our standard protocols, all bioactivity assays were carried out [13,14].

### 2.5. Effect of Neogrisphenol A on the Growth of A2780 Cells

#### 2.5.1. Cell Culture and Compound Treatment

The A2780 cells were maintained in Roswell Park Memorial Institute (RPMI) 1640 medium with 10% fetal bovine serum (FBS) and 1% Penicillin-Streptomycin Solution (100 mg/L streptomycin and 1 × 10^5^ U/L penicillin) in a 37 °C incubator chamber with 5% CO_2_. A stock solution of 2 × 10^4^ µmol/L was prepared by dissolving neogrisphenol A (abbreviated as NeoA) in DMSO.

#### 2.5.2. Cell Proliferation Assay

A2780 cells were plated in a 96-well plate with 5 × 10^3^ cells/well, and various NeoA (1.25, 2.5, 5, 10, and 20 µmol/L) concentrations were added and incubated for 24, 48, and 72 h. Each concentration consisted of five multiple pores. The number of cells and morphological changes in the cells were observed under an inverted fluorescence microscope. An amount of 20 µL of MTT (5 mg/mL) reagent was added, and the plates were incubated for 4 h in the incubator chamber. The supernatant was discarded by centrifugation, and 150 µL of DMSO was added. The absorbance was measured at 490 nm, and the inhibition rate of cell proliferation was calculated: the inhibition rate =1 (OD processing/OD control) × 100%. 

#### 2.5.3. Cell Apoptosis Assay

In order to detect the effect of NeoA on the apoptosis of A2780 cells, we selected to use the apoptosis detection kit. The A2780 cells were seeded into 6-well plates with 3 × 10^5^ cells/well and incubated with NeoA (3.0, 6.0, and 12.0 µmol/L) and 0.1% DMSO for 24 h. Cells were digested and collected with trypsin without ethylenediaminetetraacetic acid (EDTA), washed twice with phosphate-buffered saline (PBS), and mixed with 500 µL of PBS to form a cell suspension. We added 5.0 µL of annexin V-FITC and 5.0 µL of PI to the cell suspension and incubated it for 30 min in the dark. Finally, the apoptosis rate of each concentration group was assessed by flow cytometry.

#### 2.5.4. Cell Cycle Assay

RNase A and PI staining kit used the effects of NeoA in the cell cycle of A2780. The process of inoculating and treating cells is described in “Cell Apoptosis Assay”. The cells were digested and collected with trypsin-containing EDTA and fixed in 70% pre-cooled ethanol at 4 °C for more than 12 h. The cells were then washed twice with PBS, bathed in 5 µL RNase A at 37 °C for 30 min, followed by 25 µL PI solution at room temperature, and stained in the dark for 15 min. The changes in the cell cycle were analyzed by flow cytometry.

## 3. Results

### 3.1. Fungal Taxonomy

***Neohelicosporium griseum*** (Berk. and M.A. Curtis) Y.Z. Lu and K.D. Hyde, Fungal Diversity 92: 241 (2018) (Figure 1).

Index Fungorum: IF 554877; Facesoffungi number: FoF 04805.

Saprobic on decaying wood in the terrestrial habitat in a mountain. Sexual morph: undetermined. Asexual morph: Colonies on the natural substratum are superficial, gregarious, and covered with masses of crowded, glistening white conidia. Conidiophores macronematous, mononematous, erect, flexuous, cylindrical, 26–150 × 4–8 μm, ($\bar{x}$ = 88 × 6 μm, n = 20), sparsely branched, septate, arising directly on the substrate, pale brown, smooth-walled. *Conidiogenous* cells are holoblastic, mono- to polyblastic, integrated, terminal or intercalary, cylindrical, 9–14 × 5–6 μm wide ($\bar{x}$ = 12 × 5.5 μm, n = 20), with denticles that are pale brown and smooth-walled. *Conidia* solitary, acropleurogenous, helicoid, tightly coiled 1^1^/_2_ − 3^1^/_5_ times, becoming loosely coiled in water, rounded at the tip, 27–32 × 2–4.5 μm wide ($\bar{x}$ = 28.5 μm × 3.5 μm, n = 20), 119–200 μm long, multi-septate, and hyaline. 

Material examined: China, Guizhou Province, Guiyang City, Guiyang Forest Park, on decaying wood in a terrestrial habitat in the mountain, 14 September 2020, Jian Ma and Yongzhong Lu, GYSLGY1 (GZAAS 22-2002); living culture, GZCC 22-2002. GenBank accession numbers: OP470642 (ITS), OP470641 (LSU), OP698069 (RPB2), OP698068 (TEF1α).

Notes: Using morphological and phylogenetic evidence, Lu et al. [12] synonymized *Helicosporium griseum*, *H. lumbricoides,* and *H. bellus* as *Neohelicosporium griseum*. Based on the results of a multilocus phylogenetic analysis, a new isolate collected from a terrestrial habitat was determined to be *Neohelicosporium griseum* in this study (Figure 2). Moreover, *Neohelicosporium griseum* is a new record for China. We observed that the new isolate is phylogenetically distinct from the other four strains. Consequently, we examined pairwise dissimilarities of DNA sequences and discovered there are a few noticeable nucleotide differences between the new isolate and the other four known isolates in ITS and LSU sequence data (Table 1), which provides strong evidence that they belong to the same species. What must be emphasized is that the morphological characteristics of the new isolate differ from those of previously described specimens due to its larger conidia (27–32 μm diam., 2–4.5 μm wide vs. 12–15 μm diam., 1–2.5 μm wide) [15] than *Helicosporium griseum*. According to the suggestions for helicosporous hyphomycetes species identification by Lu et al. [12], the size of conidia is an essential factor in identifying helicosporous fungi. However, we cannot compare the morphology of the new isolate with the morphology of the other strains that provided molecular data due to the absence of morphological characteristics in the previously described strains. In addition, specimens with morphological characteristics are missing molecular information. It suggests that there are likely numerous incorrect taxonomic identifications within this group. Therefore, additional taxonomic research on helicosporous fungi is required.

### 3.2. Spectroscopic Data

Neogrisphenol A (**1**): earthy yellow powder; [*α*${]}_{D}^{28}$ +510 (*c* 4.0, MeOH); UV (MeOH) λ_max_ (log ε) 221 (4.42), 280 (3.13), 339 (3.66) nm; ECD (*c* 5.7 × 10^−4^ M, MeOH) λ_max_ (∆ε) 200 (+1.53), 211 (−2.09), 230 (+6.80), 254 (−2.16), 276 (−0.77), 314 (−3.79), 367 (+9.47) nm; IR (KBr) V_max_ 3429, 3051, 2923, 2855, 1635, 1454, 1338, 1241, 1210, 1128 cm^−1^ (Appendix A); ^1^H and ^13^C NMR, Table 2; HRESIMS at *m*/*z* 375.08386 [M + Na]^+^ (calcd. for C_20_H_16_O_6_N_a_, 375.083909).

Neogrisphenol B (**2**): brownish black solid; [*α*${]}_{D}^{28}$ +160 (*c* 0.5, MeOH); UV (MeOH) λ_max_ (log ε) 223 (3.33), 286 (2.59), 336 (2.92) nm; ECD (*c* 5.4 × 10^−4^ M, MeOH) λ_max_ (∆ε) 200 (+10.07), 238 (−6.31), 262 (+2.39), 309 (−11.16), 358 (+20.04) nm; IR (KBr) V_max_ 3403, 2924, 2855, 1614, 1458, 1383, 1245, 1169 cm^−1^ (Appendix A); ^1^H and ^13^C NMR, Table 2; HRESIMS at *m*/*z* 391.07761 [M + Na]^+^ (calcd. for C_20_H_16_O_7_N_a_, 391. 078824).

(S)-6-hydroxy-7-methoxy-3,5-dimethylisochroman-1-one (3): rufous solid; [α]28 D +52 (*c* 0.4, MeOH); UV (MeOH) λmax (logε) 213 (3.11), 243 (2.45), 266 (2.84), 285 (2.41), 305 (2.63) nm; ECD (*c* 9.0 × 10^−4^ M, MeOH) λmax (∆ε) 200 (+2.10), 215 (−1.23), 236 (+5.25), 254 (+0.53), 270 (+2.71), 286 (+0.80), 303 (+1.58) nm; IR (KBr) Vmax 3418, 2932, 1690, 1596, 1260, 1122, 1084 cm^−1^ (Appendix A); ^1^H and ^13^C NMR, Table 3; HRESIMS at m/z 245.07921 [M + Na]+ (calcd. for C12H14O4Na, 245.078430).

### 3.3. Structure Elucidation of Compounds ***1***–***7***

Compound **1** (Figure 3) was obtained as an earthy yellow powder, and its molecular formula was established as C_20_H_16_O_6_ based on HRESIMS data at 375.08386 [M + Na]^+^ (calcd. For C_20_H_16_O_6_N_a_, 375.083909), indicating 13 degrees of unsaturation. The 1D NMR data (Table 2 and Appendix A) of **1** and HSQC correlations showed signals for two 1,2,3-trisubstituted benzene rings, two carbonyls, a methylene, a methine, and two olefinic protons [6.74 (d, *J* = 10.0 Hz, 1H), 6.35 (d, *J* = 10.0 Hz, 1H)], as well as four exchangeable protons [12.19 (s, 1H), 12.17 (s, 1H), 6.10 (s, 1H), 5.52 (d, *J* = 4.2 Hz, 1H)]. The COSY correlations of H-3 with H-2a and H-4 along with the HMBC correlations of H-2a/C-1, C-3, C-4, H-2b/C-1, C-3, C-4, H-3/C-1, and H-4/C-2, C-3, C-4a, C-5, and C-8a indicated the presence of a substituted tetralone moiety (Figure 4 and Appendix A). The observed HMBC correlations from 2′-OH- to C-2′, C-3′, from H-3′ to C-1′, C-4a’, and from H-4′ to C-2′, C-4a’, C-5′, and C-8a’ implied the presence of another substituted *α*-(*2H*)-naphthalenone moiety (Figure 4). The observed HMBC correlations from 2′-OH to C-3, from H3′ to C-3, from H-2a to C-2′, and from H-3 to C-1′ and C-3′ suggested that there is a C-3~C-2′ linkage between the two *α*-naphthalenone moieties. The observed HMBC correlations from 8-OH to C-7, C-8, and C-8a and from 8′-OH to C-7′, C-8′, and C-8a’ indicated that 8-OH attached to C-8 and 8′-OH attached to C-8′, respectively. Thus, the planar structure of compound **1** was determined (Figure 1). The relative configuration of compound **1** was confirmed by analysis of the coupling constants, NOESY correlations (Figure 4), and ^13^C-NMR calculations (Figure 5). The large coupling constant of Hax-2a with Hax-3 (*J* = 13.3 Hz) indicated a Jaa coupling between Hax-2a and Hax-3, and a Jae coupling between Hax-3 and Heq-2b (*J* = 3.9 Hz). The very small coupling constant of Hax-3 with H-4 (*J* = 2.0 Hz) indicated a Jae coupling between them. Thus, H-3 and 4-OH were in the opposite orientations of the substituted tetralone moiety plane. The NOESY correlation of H-2a with 4-OH further confirmed this deduction. However, since the C-3~C-2′ bond can rotate, the relative configuration of the right part of compound **1** cannot be determined. Thus, we performed theoretical NMR chemical shift calculations for the two diastereomers **1**a and **1**b of **1** using the gauge-independent atomic orbital (GIAO) ^13^C NMR calculations [16]. The calculated ^13^C NMR chemical shifts of 1b showed the best agreement with the experimental values (Figure 5). Furthermore, DP4+ analysis predicted that **1**b was the most likely candidate with 100% probability (Appendix A). ECD calculations were used to determine the absolute configuration of **1**, and they revealed that the experimental and calculated ECDs for (3*R*,4*R*,2′*S*)-1 are in agreement (Figure 6). Thus, compound **1** was identified as (3*R*,4*R*,2′*S*)-1 and named neogrisphenol A.

Compound **2** (Figure 3) was obtained as a brownish-black solid, and its molecular formula was established as C_20_H_16_O_7_ based on HRESIMS data at 391.07761 [M + Na]^+^ (calcd. For C_20_H_16_O_7_N_a_, 391. 078824), indicating 13 degrees of unsaturation. The molecular formula of compound **2** has only one more oxygen atom than compound **1**, and the ^1^H-NMR spectrum is very similar, except for one less aromatic proton and one more exchangeable proton, suggesting that compound **2** has only one more phenolic hydroxyl group (10.94, s, 1H) than compound **1**. The 1D and 2D NMR data of the left part of compound 2 are almost identical to those of compound **1**, indicating they share the same planar structure and relative configuration of the left part (Appendix A). The ^1^H-NMR data of compound **2** showed the two protons [6.31, (d, *J* = 2.2 Hz, 1H), 6.17, (d, *J* = 2.2 Hz, 1H)] were meta-coupling on the right benzene ring. The HSQC correlations of H-5′/C-5′, H-7′/C-7′, along with the HMBC correlations of H-5′/C-4′, C-7′, C-8a’, H-7′/ C-5, C-6, C-8, 6′-OH/C-5′, C-7′, 8′-OH/ C-7′, C-8′, C-8a’, and H-4/C-2, C-3, C-4a, C-5, and C-8a, indicated that 6′-OH was connected to C-6′ and 8′-OH connected to C-8′, respectively. The position of the carbon-carbon double bond (C3′~4′) in the right cyclohexanone ring is also the same as in compound **1**, based on similar HMBC correlations (Appendix A). In addition, the HMBC correlations (Appendix A), similar to compound 1, proved the connection position is also a C-3~C-2′ linkage between the two naphthalenone derivative moieties. Thus, the planar structure of compound **2** was determined (Figure 3). In the same way, the relative configuration of compound **2** is defined as **2**b (Figure 7 and Appendix A). The relative configurations of compound **1** and compound **2** were assigned the same by ^13^C NMR calculation (Figure 7), and their experimental ECD curves were similar (Figure 6). The sign of the specific rotation is also positive, so the absolute configuration of compound **2** was defined as (3*R*,4*R*,2′*S*)-2 and named neogrisphenol B.

Compound **3** (Figure 1) was obtained as a rufous solid, and its molecular formula was established as C_20_H_16_O_7_ based on the HRESIMS data at 245.07921 [M + Na]^+^ (calcd. for C_12_H_14_O_4_N_a_, 245.078430), indicating 6 degrees of unsaturation. The 1D-NMR data and HSQC correlation showed signals for an aromatic proton, an exchangeable proton, a methoxyl, two methyls, one methylene, and an oxygenated methine. These data were close to those of the isochroman-1-one skeleton [17], with the exception that an aromatic proton signal was absent and one sp^3^-methoxyl signal (*δ*_H/C_ 1.98/10.6) and one oxygen-bearing sp^3^-methoxyl signal (*δ*_H/C_ 3.71/55.4) were present. The structure of the right part was confirmed by the COSY and HMBC correlations (Figure 4, Appendix A). The connection positions of the phenolic hydroxyl, methoxyl, and methyl groups on the benzene ring were determined by the key HMBC correlation from H-4a to C-5, H-8 to C-4a, C-6, C-8a, H-11 to C-7, and H-10 to C-6 (Figure 4 and Appendix A). Thus, the planar structure of compound 3 was determined. The relative configuration of compound **3** was confirmed by coupling constants. The large coupling constant of Hax-4a with Hax-3 (*J* = 12.4 Hz) indicated a Jaa coupling between Hax-4a and Hax-3, and a Jae coupling between Hax-3 and Heq-4b (*J* = 2.8 Hz). ECD calculations were used to determine compound **3**′s absolute configuration, and the results showed that the experimental and computed ECDs for (3*S*)-3 are consistent (Figure 8). Thus, compound **3** was identified as (*S*)-6-hydroxy-7-methoxy-3,5-dimethylisochroman-1-one.

Compound **4** (Figure 1), C_9_H_10_O_4_, was a white solid, ESIMS at *m*/*z* 181.2 [M-H]- (calcd. for C_9_H_10_O_4_, 182.1), 363.1 [2M − H]^−^, 183.1 [M+H]^+^, 205.1 [M + Na]^+^ (calcd. for C_9_H_10_O_4_Na, 205.1); ^1^H NMR (600 MHz, DMSO-*d*_6_) *δ* 10.69 (s, 1H), 6.16 (t, *J* = 2.1 Hz, 2H), 3.79 (s, 3H), 2.27 (s, 3H); ^13^C NMR (150 MHz, DMSO) *δ* 170.2, 161.4, 161.0, 140.7, 110.2, 107.6, 100.5, 51.8, 22.1. Compound **4** was identified as methyl orsellinate [18].

Compound **5** (Figure 1), C_14_H_14_O_3_, was a brown oil; ^1^H NMR (600 MHz, DMSO-*d*_6_) *δ* 9.44 (s, 2H), 6.34 (ddd, *J* = 2.2, 1.5, 0.8 Hz, 2H), 6.24 (ddd, *J* = 2.2, 1.4, 0.7 Hz, 2H), 6.15 (td, *J* = 2.2, 2.2, 0.6 Hz 2H), 2.18 (s, 6H); ^13^C NMR (150 MHz, DMSO) *δ* 158.5, 157.6, 140.1, 111.2, 110.1, 103.0, 21.1; ESIMS at *m*/*z* 229.1 [M − H]^−^, 459.1 [2M − H]^−^ (calcd. for C_14_H_14_O_3_, 230.1). Compound **5** was identified as diorcinol [19,20].

Compound **6** (Figure 1), C_11_H_10_O_4_, was a white needle crystal; ^1^H NMR (600 MHz, DMSO-*d*_6_) *δ* 10.67 (s, 1H), 6.42 (d, *J* = 2.2 Hz, 1H), 6.32 (d, *J* = 2.2 Hz, 1H), 6.26 (d, *J* = 1.1 Hz, 1H), 3.80 (s, 3H), 2.12 (d, *J* = 1.0 Hz, 3H); ^13^C NMR (150 MHz, DMSO) *δ* 164.2, 163.2, 157.9, 154.7, 141.8, 103.1, 102.4, 100.3, 98.6, 55.8, 18.9; ESIMS at *m*/*z* 205.1 [M-H]^−^ (calcd. for C_11_H_10_O_4_, 206.1), 229.1 [M + Na]^+^ (calcd. for C_11_H_10_O_4_Na, 229.1), 435.0 [2M + Na]^+^ (calcd. for C_22_H_20_O_8_Na, 435.0). Compound **6** was identified as 6-hydroxy-8-methoxy-3-methyl-isocoumarin [21].

Compound **7** (Figure 1), C_28_H_44_O, was a white needle crystal; ^1^H NMR (600 MHz, CDCl_3_-*d*) *δ* 5.57 (dd, *J* = 5.7, 2.6 Hz, 1H), 5.39 (dt, *J* = 5.6, 2.8 Hz, 1H), 5.20 (qd, *J* = 15.3, 7.6 Hz, 2H), 3.64 (tt, *J* = 11.2, 4.2 Hz, 1H), 2.48 (dd, *J* = 4.7, 2.4 Hz, 1H), 2.28 (ddd, *J* = 14.0, 11.7, 2.3 Hz, 1H), 2.05 (dddd, *J* = 14.4, 13.2, 7.1, 4.7 Hz, 2H), 2.00–1.94 (m, 1H), 1.93–1.86 (m, 3H), 1.85 (dd, *J* = 7.0, 5.8 Hz, 1H), 1.81–1.56 (m, 4H), 1.54–1.43 (m, 2H), 1.38 (qd, *J* = 11.0, 10.6, 5.1 Hz, 1H), 1.34–1.31 (m, 1H), 1.31–1.29 (m, 1H), 1.29–1.22 (m, 3H), 1.04 (d, *J* = 6.7 Hz, 3H), 0.96–0.89 (m, 6H), 0.83 (dd, *J* = 9.1, 6.8 Hz, 6H), 0.63 (s, 3H); ^13^C NMR (150 MHz, CDCl_3_) *δ* 141.5, 139.9, 135.7, 132.1, 119.7, 116.4, 70.6, 55.9, 54.7, 46.4, 43.0, 43.0, 41.0, 40.6, 39.2, 38.5, 37.2, 33.2, 32.2, 28.4, 23.2, 21.3, 21.2, 20.1, 19.8, 17.8, 16.4, 12.2. Compound **7** was identified as ergosterin [22].

### 3.4. Antimicrobial and Cytotoxic Activities of Compounds ***1***–***7***

The ethyl acetate (EtOAc) extracts from the rice fermentation product of *Neohelicosporium griseum* showed cytotoxicity against PC3 cell lines, with IC_50_ values of 0.4 mg/mL, as part of our ongoing research into the medical applications of helicosporous hyphomycetes. To investigate the bioactivities of compounds **1**–**7**, their antimicrobial and cytotoxic properties were evaluated. Compounds **1**–**2** and **5** exhibited moderate antibacterial activity against *Bacillus subtilis*, *Clostridium perfringens*, *Staphylococcus aureus*, and *Staphylococcus aureus* with MIC values between 16 and 31 µg/mL (Table 4). Only compound **5** exhibited antifungal activity against *Sclerotinia sclerotiorum* and *Phytophthora nicotianae var. nicotianae*, with respective IC_50_ values of 88.14 ± 2.21 µg/mL and 52.36 ± 1.38 µg/mL (Table 5). Compound **1** showed significant cytotoxicity against the A2780, PC-3, and MBA-MD-231 cell lines, with respective IC_50_ values of 3.20, 10.68, and 16.30 µM (Table 6). Compound **2** showed significant cytotoxicity against A2780 cell lines, with an IC_50_ value of 10.13 µM (Table 6).

### 3.5. Effect of Neogrisphenol A on the Growth of A2780 Cells

#### 3.5.1. Effect of Neogrisphenol A on the Cell Viability of A2780 Cells

The cytotoxicity of NeoA toward cancer cells was evaluated using the MTT technique. According to Figure 9B, the growth inhibition rate of the A2780 cells increased dose-dependently with rising NeoA concentrations and peaked by 80% at 10 mol/L. As Figure 9C shows, the quantitative analysis of the IC50 values at 24 h (4.78 ±1.57 mol/L), 48 h (3.46 ± 1.19 mol/L), and 72 h (4.74 ± 0.39 mol/L) demonstrates that the inhibitory effect of NeoA on the A2780 cells was not time-dependent. Hence, it was decided to treat A2780 cells with NeoA for 24 hours in order to conduct further research. After 24 h of A2780 cell culture, dose-dependent inhibition of cell proliferation and stimulation of cell division by NeoA were observed (Figure 9D). This study suggests that NeoA may inhibit the proliferation of A2780 cells and cause apoptosis.

#### 3.5.2. Effect of Neogrisphenol A on Apoptosis of A2780 Cells

Flow cytometry analysis showed that the number of apoptotic cells increased with the increase in NeoA concentration (Figure 9E). As shown in Figure 9G, the A2780 cells apoptosis rate increased with the increase of NeoA concentration in a concentration-dependent manner; the early and late apoptosis rates of all concentration groups differed significantly from those in the control group (DMSO), except for the low concentration group. The late apoptosis rate reached 47.80 ± 1.49% at 12 µmol/L, indicating that apoptosis induction mainly occurred in late apoptosis. This suggested that NeoA could prevent A2780 cells from proliferating by triggering early and late apoptosis.

#### 3.5.3. Effects of Different Concentrations of Neogrisphenol A on the Cycle of A2780 Cells

The percentage of cells in the G0/G1 phase dramatically dropped as NeoA concentration increased, whereas the number of cells in the S-phase sharply increased (Figure 9F). The results of the quantitative study (Figure 9I) showed that NeoA interfered with the regulation of components that control cell cycle progression and prevented the entry of the cell cycle into the S-phase. This was shown by the increase in the number of S-phase cells with rising NeoA concentrations in comparison to the control cells (DMSO). The reduction in G0/G1 phase cells indicates that NeoA probably prevented the A2780 cells from replicating their DNA. NeoA was shown to regulate the G0/G1 and S-phases of the cell cycle to inhibit the malignant proliferation of A2780 cells.

## 4. Discussion

Derivatives of the *α*-naphthalenone dimer are primarily derived from microbial metabolites. Most of them were isolated from the fungal genus *Cladosporium*, named cladosporols [23], and to a lesser extent from plants, named naphthoquinones [24,25], through the carbon-carbon bond polymerization of two *α*-naphthalenone derivatives into a dimer. Typically, they are connected through the polymerization of the benzene ring of one α-naphthalone derivative with the benzene ring of another α-naphthalone derivative or the benzene ring of one α-naphthalone derivative with the cyclohexanone ring of another α-naphthalone derivative. The biological activities of *α*-naphthalenone dimer derivatives include anticancer, antimicrobial, and so on [26,27,28,29,30,31,32]. Two novel *α*-naphthalenone dimer derivatives were isolated and identified during the study of secondary metabolites of *Neohelicosporium griseum*. They have a rare 3~2′ polymerization mode between the cyclohexanone ring of one α-naphthalone derivative and the cyclohexanone ring of another α-naphthalone derivative. This study demonstrated that helicosporous hyphomycetes are worthy of study, and future research may uncover additional active secondary metabolites with novel structures. Neogrisphenol A is anticipated to be a new anticancer drug with a bright future in both development and application.

## 5. Conclusions

Neogrisphenol A-B (**1**–**2**) and (*R*)-6-hydroxy-7-methoxy-3,5-dimethylisochroman-1-one (**3**) are three new polyketone derivatives isolated from the rice fermentation product of a new record fungus for China viz *Neohelicosporium griseum*, which is a new record for China. NOE and ^13^C-NMR calculations determined the relative configurations of compounds **1**–**2**. The absolute configurations of compounds **1**–**2** were determined through ECD calculations. Neogrisphenol A showed extensive antibacterial activity against *Bacillus subtilis*, *Clostridium perfringens*, *Staphylococcus aureus*, MRSA-strain *Staphylococcus aureus*, and *Ralstonia solanacearum*. Meanwhile, neogrisphenol B exhibited antibacterial activity against the MRSA-strain *Staphylococcus aureus*. Diorcinol showed antifungal and antibacterial activity against *Phytophthora nicotianae* var. *nicotianae* and *Sclerotinia sclerotiorum* and weak antibacterial activity against *Clostridium perfringens*. Neogrisphenol A also showed potent inhibition on A2780, PC-3, and MBA-MD-231 cell lines, particularly A2780 cell lines, with more potency than the positive drug cis-platinum. Neogrisphenol B also exhibited cytotoxicity against A2780. Neogrisphenol A dramatically reduced the rate of proliferation of A2780 cells, induced apoptosis, and prevented the S-phase of the cell cycle in a concentration-dependent manner. It is anticipated that neogrisphenol A will be further developed and utilized as a novel anticancer chemical entity.

## Figures and Tables

**Figure 1 metabolites-13-00435-f001:**
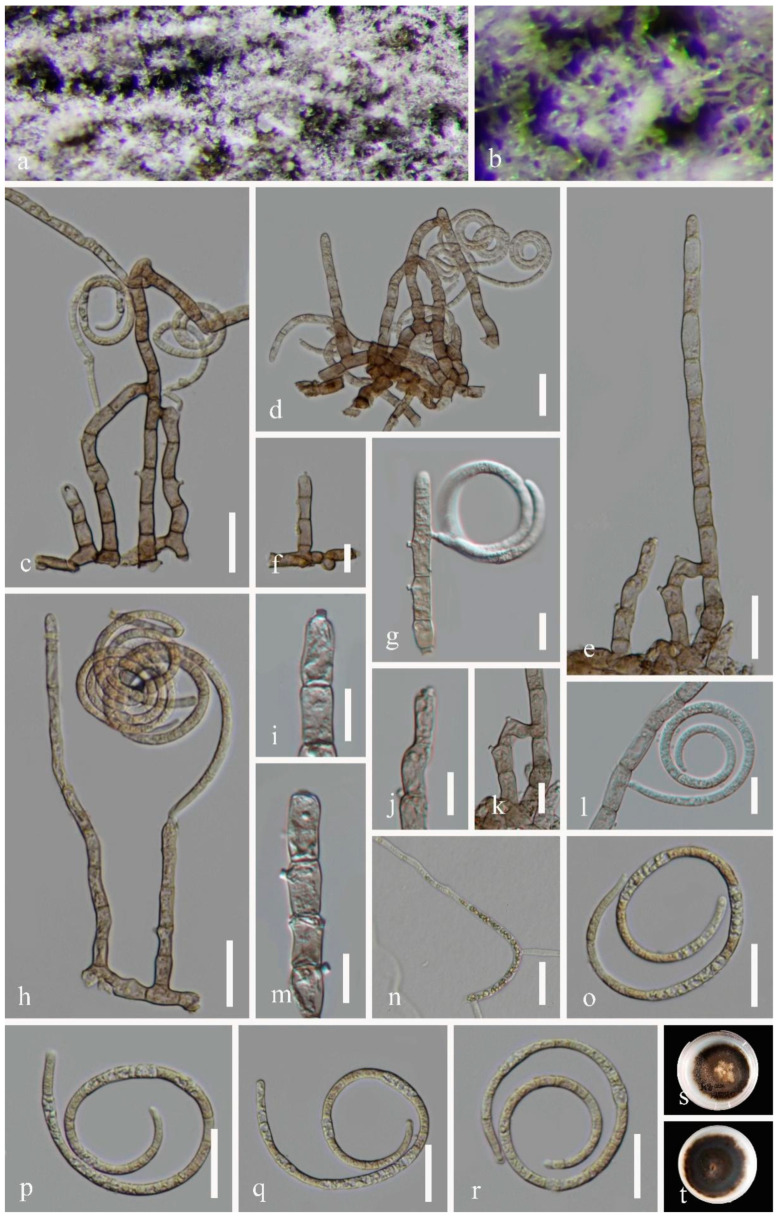
*Neohelicosporium griseum* (GZAAS 22-2002) (**a**,**b**) Colonies on natural substrates. (**c**–**f**,**h**) Conidiophores and conidia. (**g**,**i**–**m**) Conidiogenous cells and conidia. (**o**–**r**) Conidia. (**n**) Germinated conidium. (**s**,**t**) Culture on a PDA from above and reverse. Scale bars: (**c**–**e**,**h**,**n**–**r**) 20 μm, (**f**–**g**,**i**–**m**) 10 μm.

**Figure 2 metabolites-13-00435-f002:**
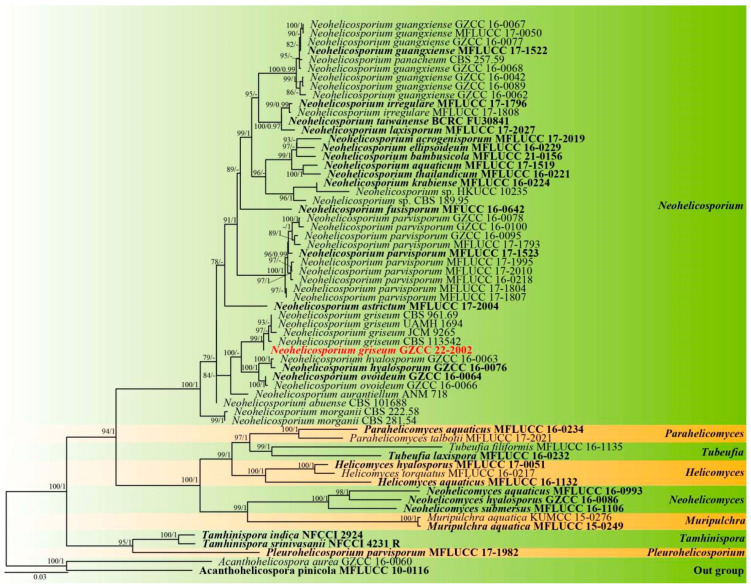
Phylogenetic tree generated from IQ-Tree v.2 analysis based on analysis of a combined set of LSU, ITS, TEF1α, and RPB2 sequence data. Bootstrap support values of maximum likelihood (ML) equal to or greater than 75% and Bayesian posterior probabilities (PP) equal to or greater than 0.95 are given near the nodes as ML/PP. *Acanthohelicospora aurea* (GZCC 16-0060) and *A. pinicola* (MFLUCC 10-0116) were used as outgroup taxa. The newly generated sequence is indicated in bold red.

**Figure 3 metabolites-13-00435-f003:**
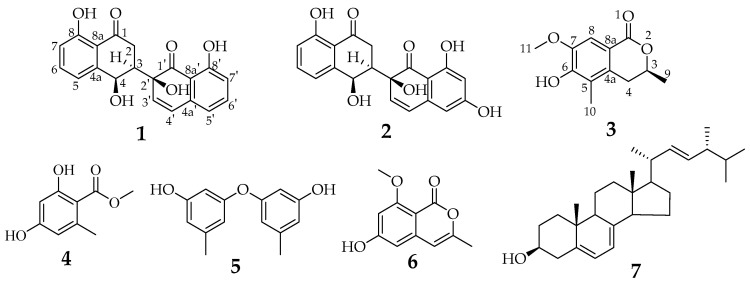
Chemical structures of compounds **1**–**7** isolated from *Neohelicosporium griseum*.

**Figure 4 metabolites-13-00435-f004:**
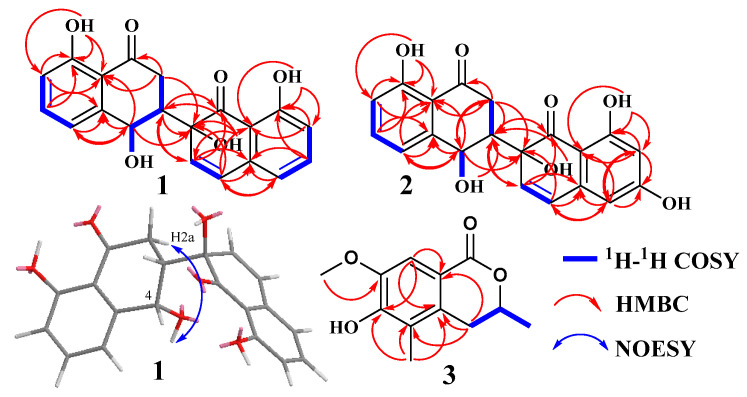
Key COSY, Key NOSEY, and HMBC correlations of compounds **1**–**3**.

**Figure 5 metabolites-13-00435-f005:**
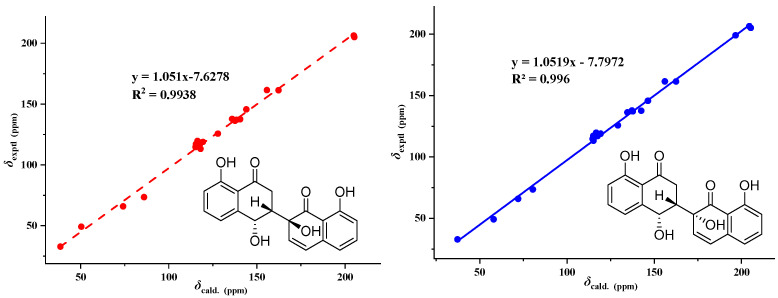
Linear regression analysis between experimental and calculated ^13^C NMR chemical shifts of isomers of **1**.

**Figure 6 metabolites-13-00435-f006:**
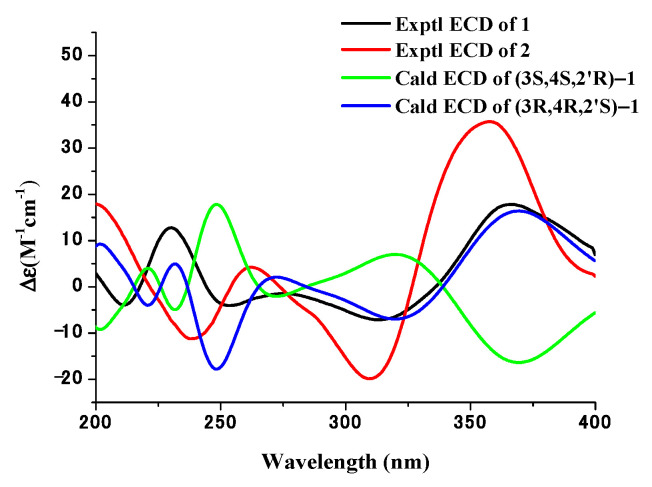
Calculated and experimental ECD spectra of compounds **1** and **2**.

**Figure 7 metabolites-13-00435-f007:**
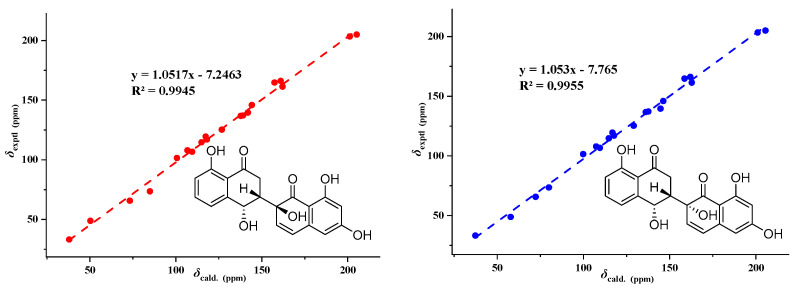
Linear regression analysis between experimental and calculated ^13^C NMR chemical shifts of isomers of **2**.

**Figure 8 metabolites-13-00435-f008:**
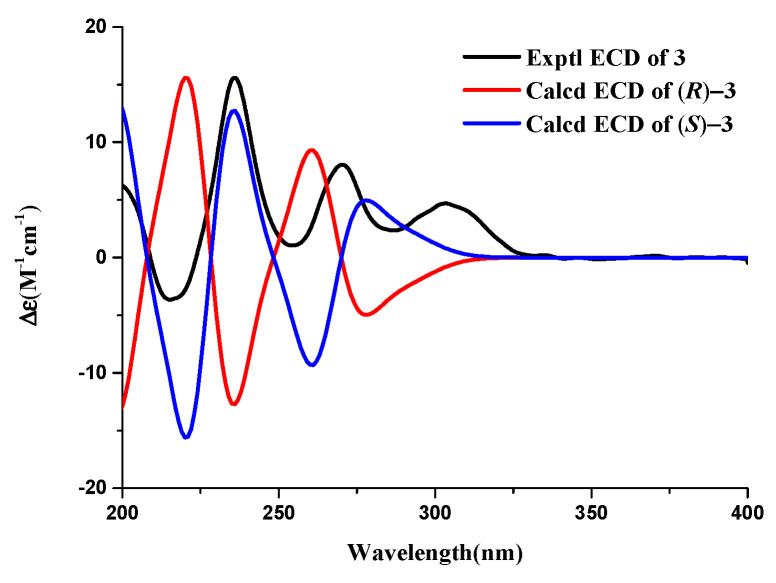
Calculated and experimental ECD spectra of compound **3**.

**Figure 9 metabolites-13-00435-f009:**
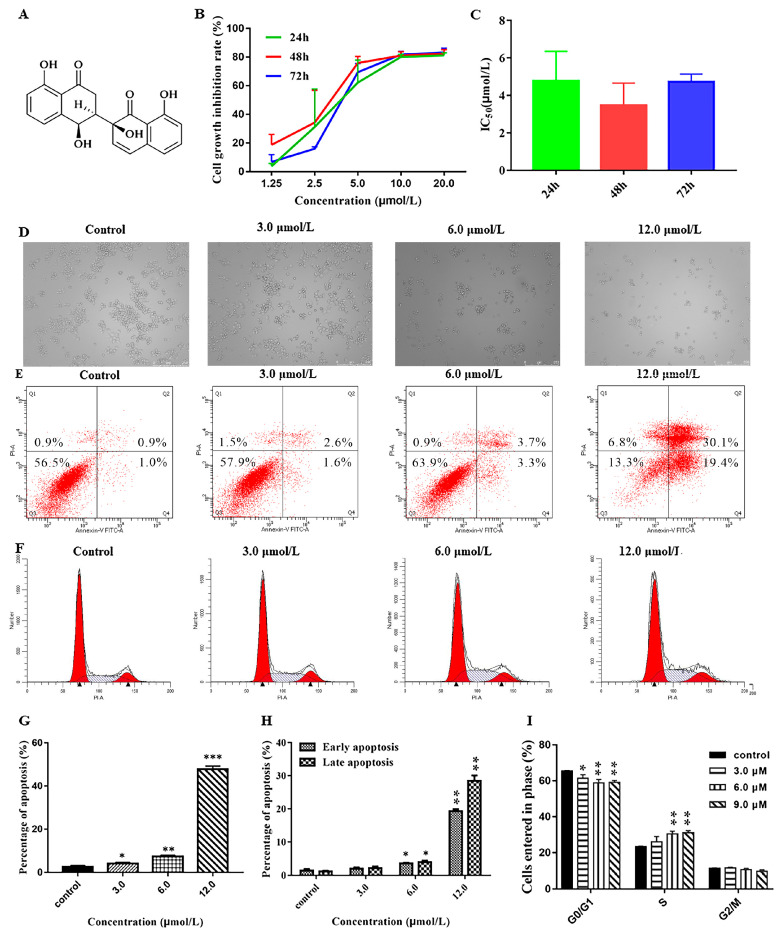
The effect of Neogrisphenol A on A2780 cell growth at various doses. (**A**) The chemical structure of NeoA; (**B**) The MTT test was used to detect the inhibitory impact of NeoA on the proliferation of A2780 cells; (**C**) The semi-inhibitory concentration (IC_50_) values of NeoA at various periods (24, 48, and 72 h); (**D**) Using an inverted microscope (magnification = 100×), morphological alterations in A2780 cells treated with various doses of NeoA for 24 h were detected; (**E**) Flow cytometry was used to assess the apoptosis of A2780 cells; (**F**) Flow cytometry was used to detect cell cycle alterations in the A2780 cell line in the control (DMSO) and NeoA groups at various doses; (**G**) The total rate of apoptosis in A2780 cells that were treated with various concentrations of NeoA; (**H**) A comparison of the effects of various concentrations of NeoA on the early and late apoptosis rates in A2780 cells; (**I**) The proportion of A2780 cells in each of the three cell-cycle stages altered following treatment with NeoA. The data is presented as the mean ± SD (n = 3). * *p* < 0.05, ** *p* < 0.01, *** *p* < 0.001 vs. the control (DMSO) group; n.s, non-significant. For statistical analysis, one-way ANOVA and multiple t tests were both performed.

**Table 1 metabolites-13-00435-t001:** Nucleotide differences between the new isolate and the other four known isolates of *Helicosporium griseum* in the ITS and LSU regions.

Strains No.	ITS Sequence			LSU Sequence		
	Position 410	Position 458	Position 460	Position 7	Position 161	Position 496
GZCC 22-2002	T	G	G	C	C	C
CBS 113542	T	T	A	C	C	T
CBS 961.69	T	G	A	T	C	T
JCM 9265	C	G	A	-	T	T
UAMH 1694	C	G	A	T	C	T

**Table 2 metabolites-13-00435-t002:** The ^1^H (600 MHz) and ^13^C NMR (150 MHz) Data of Compounds **1** and **2** in DMSO-*d*_6_.

	1		2	
No	δ_C_, Type	δ_H_ (*J* in Hz)	δ_C_, Type	δ_H_ (*J* in Hz)
1	205.1, C		205.0, C	
2	32.8, CH_2_	3.14, dd (17.7, 13.3), Ha 2.49, dd (17.7, 3.9), Hb	33.2, CH_2_	3.03, dd (17.6, 13.4), Ha 2.25, dd (17.6, 3.8), Hb
3	49.2, CH	2.61, ddd (13.3, 3.9, 2.0)	48.8, CH	2.57, ddd (13.4, 3.8, 2.0)
4	65.9, CH	4.87, d (4.2)	65.7, CH	4.98, d (4.9)
4a	145.7, C		146.0, C	
5	119.7, CH	6.88, overlap	119.5, CH	6.89, overlap
6	137.1, CH	7.51, dd (8.4, 7.4)	137.2, CH	7.53, dd (7.9, 7.9)
7	117.1, CH	6.88, overlap	117.0, CH	6.89, overlap
8	161.4, C		161.4, C	
8a	114.7, C		114.7, C	
1′	206.4, C		203.4, C	
2′	73.5, C		73.6, C	
3′	136.3, CH	6.35, d (10.0)	136.8, CH	6.43, d (10.0)
4′	125.7, CH	6.74, d (10.0)	125.3, CH	6.61, d (10.0)
4a’	137.6, C		139.6, C	
5′	118.9, CH	6.88, overlap	107.9, CH	6.31, d (2.2)
6′	137.8, CH	7.56, dd (8.4, 7.4)	166.2, C	
7′	117.1, CH	6.88, overlap	101.5, CH	6.17, d (2.2)
8′	161.5, C		164.8, C	
8a’	113.2, C		106.7, C	
4-OH		5.52, br d (4.2)		5.50, br d (4.9)
8-OH		12.17, s		12.12, s
2′-OH		6.10, br s		6.03, br s
6′-OH				10.94, s
8′-OH		12.19, s		12.68, s

**Table 3 metabolites-13-00435-t003:** The ^1^H (600 MHz) and ^13^C NMR (150 MHz) data for compound **3** in DMSO-*d*_6_.

	3	
No	δ_C_, Type	δ_H_ (*J* in Hz)
1	161.8, C	
2		
3	72.3, CH	4.36, dqd (12.0, 6.2, 2.8)
4	32.8, CH_2_	2.54, dd (16.5, 12.0), Ha 2.93, dd (16.5, 2.8), Hb
4a	112.6, C	
5	142.0, C	
6	160.7, C	
7	160.5, C	
8	97.8, CH	6.47, s
8a	104.8, C	
9	20.5, CH_3_	1.35, d (6.2)
10	10.6, CH_3_	1.98, s
11	55.4, OCH_3_	3.71, s
6-OH		10.41, br s

**Table 4 metabolites-13-00435-t004:** Minimum inhibitory concentrations (MIC, µg/mL) of **1**–**2** and **5** against bacterial test organisms.

Compound	*B*. *subtilis*	*C*. *perfringens*	*R. solanacarum*	*MRSA S* *tain ATCC43300*	*S*. *aureus*
**1**	15.63	31.25	15.63	31.25	31.25
**2**	-	-	-	31.25	-
**5**	-	250.00	-	-	-
ciprofloxacin	0.63	0.08	0.32	0.32	0.08

**Table 5 metabolites-13-00435-t005:** Antifungal activity of compound **5** against plant pathogenic fungi.

Compound	*P. nicotianae var. nicotianae*	*S. asclerotiorum*
	IC_50_ (µg/mL)	
**5**	52.36 ± 1.38	88.14 ± 2.21

**Table 6 metabolites-13-00435-t006:** Cytotoxicity of **1**–**7** against mammalian cell lines [half maximal inhibitory concentration (IC_50_): µM].

Compound	A2780	PC-3	MBA-MD-231
**1**	3.20	10.68	16.30
**2**	10.13	>20	>20
**3**	>20	>20	>20
**4**	>20	>20	>20
**5**	>20	>20	>20
**6**	>20	>20	>20
**7**	>20	>20	>20
cis-platinum	9.34	7.53	-
adriamycin	-	-	12.71

## Data Availability

The data presented in this study are available in the article and Appendix A.

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
