# Peer review of "Neogrisphenol A, a Potential Ovarian Cancer Inhibitor from a New Record Fungus Neohelicosporium griseum"

_metabolites, 2023, doi:10.3390/metabo13030435_

Round 1

Reviewer 1 Report

This review concerns the article type manuscript, entitled “Neogrisphenol A, a potential ovarian cancer inhibitor from a new rercord fungus Neohelicosporium griseum”, and submitted to Metabolites journal (Manuscript ID: metabolites-2260186).

Due to the area of my expertise my opinion can be valid only for structure determination. Three (3) new compounds were isolated and their structures were determined using NMR (1H, 13C, HSQC, COSY, NOESY, HMBC), MS (ESIMS, HR-ESIMS), and FTIR techniques. The spectra for the compounds 1-3 are presented in Supplementary. The stereochemistry was evaluated using calculated and experimental ECD spectra. Basically, the structure determination is correct. The data are well presented and clear. Therefore, in my opinion, the work should be accepted for publication.

Minor remarks.

The figures 1, 16, and 24 in supplementary are somehow incomplete. Please, correct.

The word “rercord” in the title.

Author Response

Dear Reviewer

Thank you very much for your comments and suggestions. I have revised the manuscript according to your comments and suggestions. The following are the responses.

1.The figures 1, 16, and 24 in supplementary are somehow incomplete. Please, correct.

Response 1: I'm sorry that I didn't pay attention to the details, but I have reprocessed all the one-dimensional NMR spectra. But the mass spectra, I don't understand where somehow incomplete. Do you mean the m/z should start at 0? If that is the case, I have asked the staff who helped me test the mass spectrum to help me produce the report again, but unfortunately, the previous data has been cleaned.

2.The word “rercord” in the title.

Response 2: I'm sorry I made a mistake here. It should be "record". I have corrected it.

Reviewer 2 Report

The manuscript by Li-Juan Zhang etal. Entitled: Neogrisphenol A, a potential ovarian cancer inhibitor from a new record fungus Neohelicosporium griseum, presents the identification and characterization of cytotoxicity and apoptosis induction on cancer cells of a series of new natural products: two bisnaphthalenone-derived compounds and isocoumarins. This study is comprehensive going from a thorough und in-depth chemical analysis up to the biological assays of these new natural products on cancer lines. This study has also an obvious interest in the field of anticancer treatments. I have a few questions and some minor comments.

Questions

1. Page 6, line 195. The authors indicate that there is only one noticeable nucleotide difference between their new isolate of Neohelicosporium griseum and the four previous known. Could the authors indicate what nucleotide change this is and, if known, what gene is impacted by this nucleotide change?

2. Page 15. The increase of apoptotic cells with increasing concentration of NeoA as shown in Fig. 9G seems not linear. The amount of apoptotic cells goes from 0 to 8% going from 0 to 6 µM NeoA, then for an additional 6µM (going from 6 to 12 µM NeoA), the percentage of apoptotic cells goes from 8 up to 52% Could the authors comment on that nonlinearity of the response (that is in itself very interesting).

3. The authors focused only on neogrisphenol A for their cytotoxicity study. There is a 2012 paper by Tsakalozou et al. (Combination Effects of Docetaxel and Doxorubicin in Hormone-Refractory Prostate Cancer Cells. Biochem Res Int. 2012; 2012: 832059) that shows that a strong synergy was observed when PC3 cells were treated with docetaxel at concentrations lower than its IC50 values 2-times IC50 concentration of doxorubicin. Could the authors tried to check for some synergy existing between NeoA and NeoB ?

Minor comments

1. In the title the word “rercord” is absolutely unknown to me … Could it be a typo, reading “recorded” instead of “rercord”? Could the authors correct this?

2. When presenting cell lines, such as PC3, could the authors indicate precisely which cancer line it is, i.e. instead of writing simply “PC3” the authors should write: “human prostate cancer PC3 cell line”.

3. The abbreviation “Neo.A” for neogrisphenol A, with its central period, is a bit confusing for English readers, the authors should prefer using NeoA or NeoB without a period.

4. There are no abbreviation list.

Author Response

Dear Reviewer

Thank you very much for your comments and suggestions. I have revised the manuscript according to your comments and suggestions. The following are the responses.

Questions

  1. Page 6, line 195. The authors indicate that there is only one noticeable nucleotide difference between their new isolate of Neohelicosporium griseum and the four previous known. Could the authors indicate what nucleotide change this is and, if known, what gene is impacted by this nucleotide change?

Response 1: We examined pairwise dissimilarities of DNA sequences and discovered there are a few noticeable nucleotide differences between the new isolate and the other four known isolates in ITS and LSU sequence data (Table 1), which providing strong evidence that they belong to the same species. We also provided the Table 1 in the manuscript.

Table 1. Nucleotide differences between the new isolate and the other four known isolates of Helicosporium griseum in ITS and LSU regions.

Strains No.

ITS sequence

LSU sequence

Position 410

Position 458

Position 460

Position 7

Position 161

Position 496

GZCC 22-2002

T

G

G

C

C

C

CBS 113542

T

T

A

C

C

T

CBS 961.69

T

G

A

T

C

T

JCM 9265

C

G

A

-

T

T

UAMH 1694

C

G

A

T

C

T

  1. Page 15. The increase of apoptotic cells with increasing concentration of NeoA as shown in Fig. 9G seems not linear. The amount of apoptotic cells goes from 0 to 8% going from 0 to 6 µM NeoA, then for an additional 6µM (going from 6 to 12 µM NeoA), the percentage of apoptotic cells goes from 8 up to 52% Could the authors comment on that nonlinearity of the response (that is in itself very interesting).

Response 2: Thank you for raising this very good question. As shown in Figure 9G, the apoptosis rate at concentration 3.0 µmol/L is 4.03%, at 6.0 µmol/L is 7.40%, and at 12.0 µmol/L is 47.80%. This result indicates that the apoptosis rate of cells is dose-dependent with NeoA concentration, which is consistent with the report of Li et al. (2022). When the concentration changes from 6 µmol/L to 12 µmol/L, the apoptosis rate is indeed not linear, which may be caused by the large concentration span or related to the structure and efficacy of the drug, and the mechanism here needs to be further studied. In the subsequent study, we will be more detailed and in-depth.

  1. The authors focused only on neogrisphenol A for their cytotoxicity study. There is a 2012 paper by Tsakalozou et al. (Combination Effects of Docetaxel and Doxorubicin in Hormone-Refractory Prostate Cancer Cells. Biochem Res Int. 2012; 2012: 832059) that shows that a strong synergy was observed when PC3 cells were treated with docetaxel at concentrations lower than its IC50 values 2-times IC50 concentration of doxorubicin. Could the authors tried to check for some synergy existing between NeoA and NeoB?

Response 3: Thank you for asking this very good question and pointing the way for our future research. In this study, NeoA had the best inhibitory effect on A2780. Therefore, we focused on the effect of NeoA on the growth of human ovarian cancer cells A2780, and did not focus on whether there was a synergistic effect between NeoA and NeoB. However, in the follow-up study, I will conduct the corresponding study after obtaining enough NeoB according to your suggestions.

Minor comments

  1. In the title the word “rercord” is absolutely unknown to me … Could it be a typo, reading “recorded” instead of “rercord”? Could the authors correct this?

Response 1: I'm sorry I made a mistake here. It should be "record". I have corrected it.

  1. When presenting cell lines, such as PC3, could the authors indicate precisely which cancer line it is, i.e. instead of writing simply “PC3” the authors should write: “human prostate cancer PC3 cell line”.

Response 2: I have corrected it to " human prostate cancer PC3 cell line " according to your suggestion.

  1. The abbreviation “Neo.A” for neogrisphenol A, with its central period, is a bit confusing for English readers, the authors should prefer using NeoA or NeoB without a period.

Response 3: I have corrected it to " NeoA " according to your suggestion.

  1. There are no abbreviation list.

Response 4: On line 134 I have commented " abbreviated as NeoA ". This is the only acronym, so I don't have a list of abbreviations attached.

Reviewer 3 Report

This manuscript entitled “Neogrisphenol A, a potential ovarian cancer inhibitor from a new rercord fungus Neohelicosporium griseum” reported the isolation of two new polyketides (Neogrisphenol A and neogrisphenol B) and one new isochroman-1-one along with some known compounds. Compounds were elucidated using 1D, 2D NMR, 13C-NMR, HRMS and stereochemistry was assigned by ECD experiments. These compounds showed moderate antimicrobial activity while Neogrisphenol A displayed potent cytotoxicity against A2780 cancer cells in compared to cis-platinum. This manuscript may be suitable for publication in metabolites journal after below comments

1.     Typing mistakes should be corrected like rercord fungus in title, cil-platinum (line 32 and 60), add µM after value (line 33). Authors can go through whole manuscript for minor typing errors.

2.     The absolute configuration of compound 1, 2 and 3 was determined by ECD calculations. If possible, authors can try to develop a single crystal (either 1 or 2) for X-ray crystallography studies as additional evidence of absolute configuration.

3.     The stereochemistry of compound 3 (figure 1) is S, however authors mentioned on page 12 that compound 3 was identified as R isomer. The ECD spectra of compound 3 looks like for S isomer. Please clarify this.

4. On page 15, please increase the clarity of x- and y-axis of figure 9D, E, and F for clear visibility.  Authors can mention in text about percentage of apoptotic cells in each quadrant in annexin binding assay.

Author Response

Dear Reviewer

Thank you very much for your comments and suggestions. I have revised the manuscript according to your comments and suggestions. The following are the responses.

  1. Typing mistakes should be corrected like rercord fungus in title, cil-platinum (line 32 and 60), add µM after value (line 33). Authors can go through whole manuscript for minor typing errors.

Response 1: I'm sorry to make such a stupid mistake. I have corrected it as you suggested

  1. The absolute configuration of compound 1, 2 and 3 was determined by ECD calculations. If possible, authors can try to develop a single crystal (either 1 or 2) for X-ray crystallography studies as additional evidence of absolute configuration.

Response 2: Unfortunately, I tried many solvents and methods, but compounds 1 and 2 failed to grow single crystals.

  1. The stereochemistry of compound 3 (figure 1) is S, however authors mentioned on page 12 that compound 3 was identified as R isomer. The ECD spectra of compound 3 looks like for S isomer. Please clarify this.

Response 3: I'm sorry it was my fault. The ECD spectra of compound 3 has been re-fitted, and all positions involving absolute configuration have been corrected to S isomer.

  1. On page 15, please increase the clarity of x- and y-axis of figure 9D, E, and F for clear visibility. Authors can mention in text about percentage of apoptotic cells in each quadrant in annexin binding assay.

Response 4: I have adjusted their clarity according to your advice. The percentage of apoptotic cells in each quadrant is plotted directly on the graph.
